# Epidemiological shifts in pediatric respiratory pathogens in Shenzhen, China: impacts of COVID-19 control measures and relaxation

Tao Wu,[1] Liyang Zhong,[1,2] Zhenmin Ren,[1] Yunshen Chen[1]

**ABSTRACT**   This retrospective study analyzed epidemiological features of infections by 13 common respiratory pathogens among children in Shenzhen, China, from January 2020 to February 2025. A total of 73,886 throat swabs from pediatric patients with suspected respiratory infections were tested using multiplex fluorescence PCR-capillary electrophoresis. Human rhinovirus (HRV), respiratory syncytial virus (RSV), *Mycoplasma pneumoniae* (MP), and human parainfluenza virus (HPIV) were the most prevalent, with positive rates of 24.57%, 12.46%, 11.46%, and 6.86%, respectively. Pathogen positivity rates significantly differed between the coronavirus disease 2019 (COVID-19) pandemic (2020–2022) and post-relaxation period (2023–2025) ($P < 0.0001$). Seasonal patterns shifted: HPIV peaked in summer, HRV in autumn-winter, MP in summer-autumn, and RSV in spring. After control measures eased, positivity increased across all age groups. HPIV was highest in children aged 6 months to 1 year, HRV in 3–6 years, MP in >6 years, and RSV in 0–6 months. Age- and sex-related differences were significant (all $P < 0.001$). These findings indicate that evolving COVID-19 mitigation strategies have altered epidemiological patterns, highlighting the need for targeted prevention considering high-risk groups, co-infections, and seasonality.

**IMPORTANCE** This large-scale pediatric study reveals how coronavirus disease 2019 control measures and their relaxation reshaped respiratory pathogen epidemiology in Southern China. We identified major shifts in seasonal peaks, age-specific susceptibility, and overall infection prevalence, particularly for human rhinovirus, respiratory syncytial virus, *Mycoplasma pneumoniae*, and human parainfluenza virus. The narrowing or shifting of epidemic windows implies altered transmission dynamics, likely driven by nonpharmaceutical interventions and subsequent immunity gaps. The significant post-relaxation surge across all age groups underscores the vulnerability of children to multiple pathogens once restrictions ease. These findings provide crucial evidence for optimizing pediatric respiratory infection surveillance and designing adaptive, season- and age-specific prevention strategies to mitigate future outbreaks.

**KEYWORDS**    respiratory pathogens, epidemiology, COVID-19 pandemic, children

Acute respiratory tract infection (ARTI) remains one of the most prevalent infectious diseases among children worldwide, particularly those under 5 years old, constituting a major burden in pediatric outpatient and emergency settings (1, 2). ARTI is highly complex in etiology, with numerous viruses, such as influenza virus (IFV), respiratory syncytial virus (RSV), rhinovirus (HRV), adenovirus (ADV), human parainfluenza virus (PIV), and human bocavirus (BoV) involved in its transmission. The incidence of ARTI peaks during winter and seasonal transitions and spreads rapidly in densely populated environments. According to the World Health Organization, ARTI—especially

**Peer Reviewer** Karen K. Kyuregyan, Mechnikov Research Institute for Vaccines and Sera, Moscow, Russia

Address correspondence to Zhenmin Ren, yyleo79@163.com, or Yunshen Chen, chenyunshenglw@163.com.

The authors declare no conflict of interest.

lower respiratory tract infections (LRTI)—continues to rank among the highest causes of morbidity and mortality globally, significantly impacting child health and public healthcare resources (3). In recent years, the proportion of ARTI cases among children in China has notably increased. While more than 200 respiratory viruses have been identified, children remain at high risk, with viral infections occurring far more frequently than bacterial ones (4).

Since the outbreak of coronavirus disease 2019 (COVID-19) in early 2020, public health systems have been under tremendous pressure. China and other countries have implemented a range of non-pharmaceutical interventions (NPIs), such as mask-wearing, social distancing, restrictions on gatherings, improved personal hygiene, reduced mobility, and online education (5). These NPIs not only helped suppress the spread of severe acute respiratory syndrome coronavirus 2 (SARS-CoV-2) but also significantly shifted the epidemiological patterns of other common respiratory viruses like IFV, RSV, and RV (6). Studies have shown that the implementation of NPIs in countries, such as Italy, led to a marked decrease in emergency visits for respiratory infections, suggesting that public health interventions can greatly impact ARTI transmission (7).

Currently, there is a lack of systematic epidemiological analysis of pediatric ARTI during different COVID-19 prevention phases and intervention measures, particularly in mega-cities, such as Shenzhen. In this study, we retrospectively analyzed clinical data of children presenting with suspected respiratory infections in Shenzhen from January 2020 to February 2025, systematically examining the epidemiological trends and dynamic changes of ARTI and other common respiratory viral infections under differing COVID-19 pandemic stages and major interventions. The findings aim to provide scientific support for early warning, precision prevention, and healthcare management of pediatric respiratory infections in Shenzhen, as well as inform optimized urban public health strategies.

## MATERIALS AND METHODS

### Study participants

To investigate the epidemiological trends of 13 respiratory pathogens in children during the COVID-19 pandemic and following the relaxation of control measures, this study consecutively enrolled eligible patients according to the following criteria: (i) children aged 0 to 18 years who visited Shenzhen Children's Hospital between January 2020 and February 2025; (ii) patients presenting with suspected respiratory infection symptoms, such as cough, fever, rhinorrhea, or shortness of breath, for whom a throat swab specimen was collected within 48 h of symptom onset; and (iii) children with confirmed COVID-19 infection were excluded from the analysis. For each episode of illness, only the results from the first collected specimen were included for analysis. This study was conducted in accordance with the Declaration of Helsinki and was approved by the Medical Ethics Committee of Shenzhen Children's Hospital (Approval No. 202318002). Written informed consent was waived due to the retrospective design of the study.

Clinical data were obtained from the laboratory information system of Shenzhen Children's Hospital, including age, sex, patient type (inpatient or outpatient), nasopharyngeal swab collection date, and the test results (positive or negative) for 13 respiratory pathogens. Patients were stratified into six age groups: (i) 0–6 months; (ii) 7–12 months; (iii) 1–3 years; (iv) 3–6 years; (v) 6–12 years; and (vi) 12–18 years. Based on the date of sample collection, seasons were recoded as follows: spring (March–May), summer (June–August), autumn (September–November), and winter (December–February). In accordance with COVID-19 prevention and control policies in China, the study period was divided into two phases: Phase I (during the pandemic, from January 2020 to December 2022); and Phase II (after the relaxation of pandemic control measures, from January 2023 onwards).

## Sample collection

Nasopharyngeal swabs were collected by physicians with specialized training. During the procedure, a sterile swab was inserted fully into the nasal cavity and gently rubbed against the turbinates several times to obtain epithelial cells. After sampling, the swab was immersed in the extraction buffer within the collection tube and agitated thoroughly. The exterior of the tube was repeatedly pressed by hand to ensure thorough mixing of the swab with the extraction solution. The swab was then removed, and the resulting eluate served as the specimen for subsequent analysis.

## Laboratory assay

Nucleic acid extraction was performed using an automated extractor and corresponding extraction reagents. During the process, the designated volume of samples—including both positive and negative controls—was collected, and 2 µL of RT-PCR internal control was added for extraction. Pathogen detection was performed using a multiplex assay kit for 13 respiratory pathogens (fluorescence PCR–capillary electrophoresis method; Health Gene Technologies Co., Ltd., Ningbo, China). The detectable pathogens included influenza A virus (FluA, subtypes H1N1 and H3N2), influenza B virus (FluB), human parainfluenza virus (HPIV), ADV, *Mycoplasma pneumoniae* (Mp), *Chlamydophila pneumoniae* (Cp), human bocavirus (HBoV), coronavirus (CoV), HRV, human metapneumovirus (HMPV), and human RSV. Infection status was determined by comparing the peak heights of specific PCR products in each well with those of standard samples.

## Statistical analysis

Since the age distribution of patients did not follow a normal distribution, age was presented as the median and interquartile range (IQR). Continuous variables with non-normal distribution were analyzed using the Kruskal–Wallis rank-sum test. Categorical variables were compared using the $\chi^2$ test or Fisher's exact test as appropriate. All tests were two-sided, and a $P$ value of less than 0.05 was considered statistically significant. Statistical analyses were conducted using R software program, version 4.3.1.

## RESULTS

### Positivity and distribution of 13 respiratory pathogens

From January 2020 to February 2025, a total of 73,886 respiratory specimens were collected, including 31,423 during the pandemic and 42,463 after the pandemic period. The positive rates of 13 respiratory pathogens are summarized in Table 1. Among these, the four most prevalent pathogens were HRV, RSV, MP, and HPIV, with positive rates of 24.57% (18,154/73,886), 12.46% (9,208/73,886), 11.46% (8,470/73,886), and 6.86% (5,066/73,886), respectively. Overall, the detection rate of respiratory viruses after the relaxation of preventive measures was significantly higher than that during the pandemic ($\chi^2 = 1716$, $P < 0.0001$). Significant differences in positive rates between the two periods were observed for HRV ($\chi^2 = 96.21$, $P < 0.0001$), RSV ($\chi^2 = 138.5$, $P < 0.0001$), MP ($\chi^2 = 4162$, $P < 0.0001$), and HPIV ($\chi^2 = 17.76$, $P < 0.0001$). During the pandemic, HRV (21.92%) and RSV (14.10%) had the highest detection rates, whereas HRV (25.07%) and MP (17.71%) predominated after the relaxation of interventions.

With respect to mixed viral infections, for the four pathogens with the highest positive rates, most types of dual infections increased significantly after the relaxation of control measures, with HRV-associated co-infections being the most common, followed by RSV. Notably, for all 13 pathogens, the incidence of multiple viral infections rose Markedly post-relaxation—during the pandemic, 361 cases of triple infections and 18 cases of quadruple infections were identified; whereas after the relaxation of measures, 751 cases of triple infections and 54 cases of quadruple infections were recorded (Table 2).

**TABLE 1** The positive rates of 13 respiratory pathogens

| Pathogen | Total positive cases (n) | Positive rate (%) | During the COVID-19 pandemic (n = 31,423) | After the COVID-19 pandemic (n = 42,463) | $\chi^2$ | P |
|---|---|---|---|---|---|---|
| HRV | 18,154 | 24.57 | 6,889 (21.92) | 10,645 (25.07) | 96.21 | <0.0001 |
| RSV | 9,208 | 12.46 | 4,432 (14.10) | 4,786 (11.27) | 138.50 | <0.0001 |
| MP | 8,470 | 11.46 | 784 (2.49) | 7,520 (17.71) | 4,162 | <0.0001 |
| HPIV | 5,066 | 6.86 | 1,962 (6.24) | 2,997 (7.06) | 17.76 | <0.0001 |
| ADV | 4,235 | 5.73 | 964 (3.07) | 3,234 (7.62) | 690.50 | <0.0001 |
| FluA | 4,156 | 5.62 | 1,100 (3.50) | 3,038 (7.15) | 450.50 | <0.0001 |
| HMPV | 3,652 | 4.94 | 1,769 (5.63) | 1,952 (4.60) | 42.12 | <0.0001 |
| H1N1 | 1,978 | 2.68 | 63 (0.20) | 1,945 (4.58) | 1,305 | <0.0001 |
| H3N2 | 1,785 | 2.42 | 878 (2.79) | 832 (1.96) | 37.77 | <0.0001 |
| Hcov | 1,728 | 2.34 | 608 (1.93) | 1,122 (2.64) | 38.40 | <0.0001 |
| Boca | 1,282 | 1.74 | 489 (1.56) | 751 (1.77) | 4.598 | <0.05 |
| FluB | 1,100 | 1.49 | 415 (1.32) | 701 (1.65) | 12.71 | <0.001 |
| Ch | 541 | 0.73 | 198 (0.63) | 329 (0.77) | 5.108 | <0.05 |
| Total | 42,463 | 57.47 | 17,665 (56.22) | 30,167 (71.04) | 1,716 | <0.0001 |

## Seasonal distribution

Our study revealed notable shifts in the seasonal distribution of respiratory viruses during and after the COVID-19 pandemic. During the pandemic, HRV (Fig. 1A) mainly peaked in autumn and winter and maintained high levels in those seasons both during and after the pandemic, reflecting pronounced seasonality. MP (Fig. 1B) showed a relatively stable distribution throughout the year during the pandemic, with only a minor dip in spring; however, its positive rate rose sharply in summer and autumn post-pandemic. RSV (Fig. 1C) was mainly prevalent in summer and autumn during the pandemic but displayed a steady decline from spring to winter after the pandemic. HPIV (Fig. 1D) had higher positive rates in autumn and winter during the pandemic, while the seasonal peak shifted notably to summer post-pandemic, suggesting a clear alteration in its temporal prevalence. Statistical analysis of the seasonal distribution of these respiratory viruses during and after the pandemic confirmed that the differences in positivity rates among different seasons were statistically significant for each pathogen as well as for the overall data.

**TABLE 2** Changes in mixed infection rates of 13 respiratory pathogens before and after relaxation of control measures

| Type | During the COVID-19 pandemic (n = 31,423) | After the COVID-19 pandemic (n = 42,463) | $\chi^2$ | P |
|---|---|---|---|---|
| HRV+RSV | 359 (1.14) | 556 (1.31) | 4.113 | <0.05 |
| HRV+MP | 121 (0.39) | 1,708 (4.02) | 989.6 | <0.0001 |
| HRV+HPIV | 293 (0.93) | 559 (1.32) | 23.36 | <0.0001 |
| RSV+MP | 20 (0.06) | 183 (0.43) | 88.93 | <0.0001 |
| RSV+HPIV | 39 (0.12) | 140 (0.33) | 31.58 | <0.0001 |
| MP+HPIV | 20 (0.06) | 248 (0.58) | 135.3 | <0.0001 |
| HRV+RSV + MP | 3 (0.01) | 27 (0.06) | 12.98 | <0.001 |
| HRV+RSV + HPIV | 7 (0.01) | 19 (0.04) | 2.592 | 0.107 |
| HRV+MP + HPIV | 4 (0.01) | 41 (0.10) | 20.85 | <0.0001 |
| RSV+MP + HPIV | 0 | 8 (0.02) | 5.921 | <0.05 |
| HRV+RSV + MP+HPIV | 0 | 1 (0.01) | 0.740 | 0.390 |
| Double infection | 4,249 (13.52) | 5,629 (13.26) | 1.101 | 0.294 |
| Triple infection | 361 (1.15) | 751(1.77) | 46.79 | <0.0001 |
| Quadruple infection | 18 (0.06) | 54(0.13) | 9.060 | <0.01 |

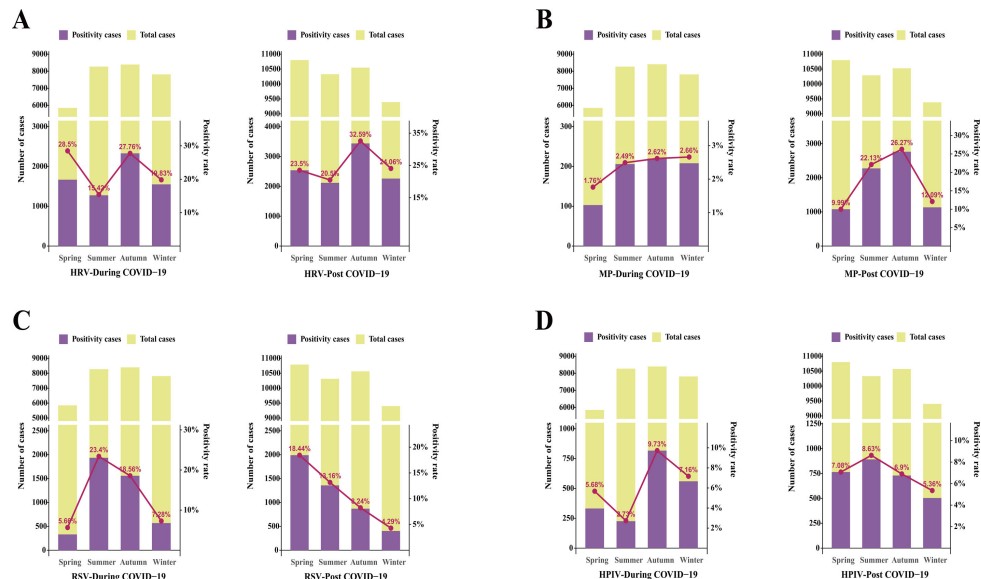

**FIG 1** Seasonal distribution patterns of respiratory viruses during and after the COVID-19 pandemic, showing notable shifts and statistically significant differences in seasonal prevalence for each pathogen. (A) HRV, (B) MP, (C) RSV, and (D) HPIV. HRV, human rhinovirus; MP, *Mycoplasma pneumoniae*; RSV, respiratory syncytial virus; HPIV, human parainfluenza virus.

## Gender distribution

Among the 73,886 patients included in the analysis, 43,455 (58.81%) were male, and 30,431 (41.19%) were female. The overall viral positive rate was 67.03% in males, notably higher than 65.41% in females. This gender difference in overall positive rates was statistically significant ($\chi^2 = 20.90$, $P < 0.0001$). Further analysis revealed significant differences in positive rates between males and females both during and after the COVID-19 pandemic ($P < 0.0001$). With regard to major pathogens, significant gender-based differences in positive rates were observed for HRV ($\chi^2 = 39.60$, $P < 0.0001$), RSV ($\chi^2 = 14.28$, $P < 0.001$), MP ($\chi^2 = 62.66$, $P < 0.0001$), and HPIV ($\chi^2 = 11.89$, $P < 0.001$). Additionally, post-pandemic data showed that the positive rates of HRV, MP, and HPIV increased markedly in both male and female cases, while the positive rate of RSV significantly decreased ($P < 0.0001$) (Fig. 2).

## Age distribution

The patients were divided into six age groups: (i) 0–6 months, (ii) 7–12 months, (iii) 1–3 years, (iv) 3–6 years, (v) 6–12 years, and (vi) 12–18 years. Across different age groups, the positivity rates of four major respiratory pathogens during and after the COVID-19 pandemic were compared (Table 3). Analysis of respiratory pathogen detection rates during and after the COVID-19 pandemic showed a significant increase in overall positive rates across all age groups following the relaxation of public health measures (Fig. 3). Specifically, the highest HPIV positivity rate was observed in children aged 6 months to 1 year, increasing from 8.61% during the pandemic to 12.16% after restrictions were lifted. In the 3–6 years group, HRV had the highest positivity rate, rising from 26.31% to 29.02%. Among individuals older than 6 years, the MP positivity rate showed a substantial increase from 5.09% during the pandemic to 28.33% post-relaxation. The highest RSV positivity rate was seen in infants aged 0–6 months, with rates of 24.21% during the pandemic and 25.32% after the easing of restrictions. Overall, the differences in viral infection rates among the various age groups were statistically significant ($\chi^2 = 1530$, $P < 0.0001$). Further analysis of individual pathogens revealed significant variation in positivity rates across age groups for HRV ($\chi^2 = 546.8$, $P < 0.0001$), RSV ($\chi^2 = 4457$, $P < 0.0001$), MP ($\chi^2 = 3721$, $P < 0.0001$), and HPIV ($\chi^2 = 746.5$, $P < 0.0001$) (Fig. 4).

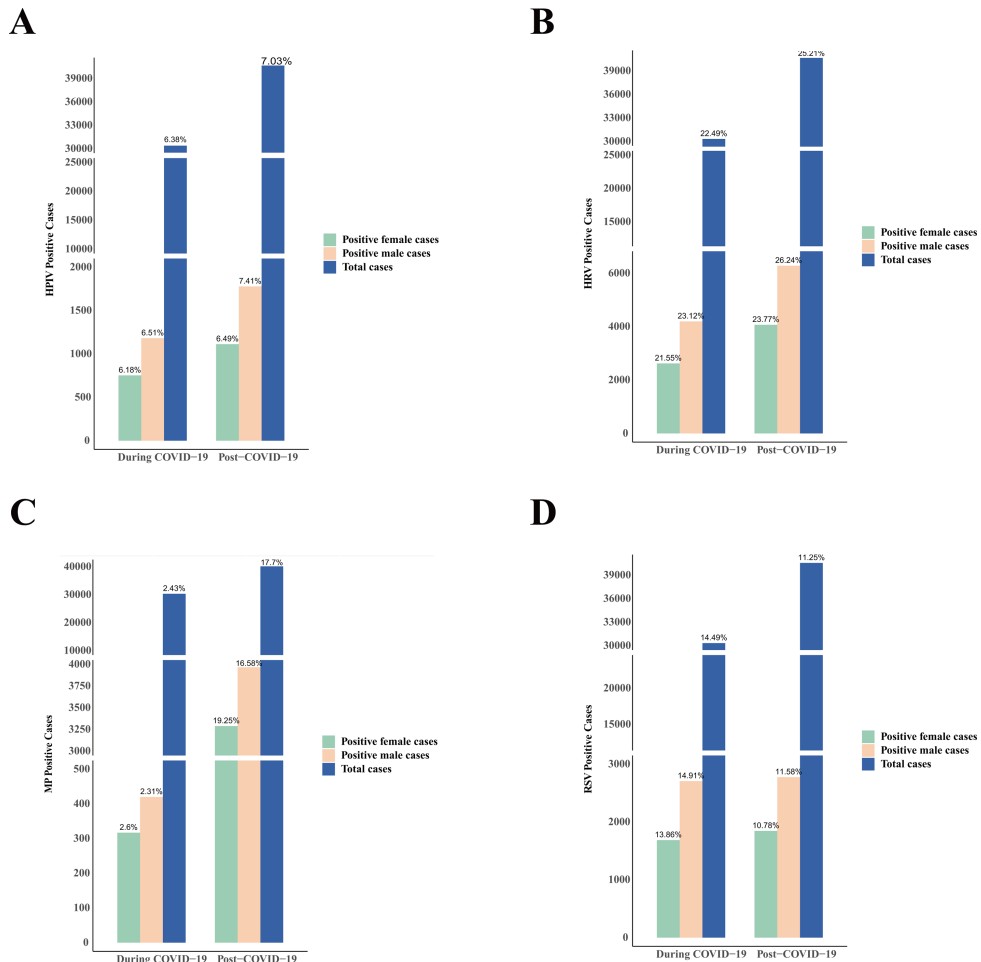

**FIG 2** The number and percentage of virus-positive specimens by gender during and after the COVID-19 pandemic, including (A) HRV, (B) MP, (C) RSV, and (D) HPIV. HRV, human rhinovirus; MP, *Mycoplasma pneumoniae*; RSV, respiratory syncytial virus; HPIV, human parainfluenza virus.

## DISCUSSION

ARTIs remain a leading cause of morbidity and mortality among children worldwide, accounting for millions of pediatric deaths annually. Since the outbreak of COVID-19, a series of stringent public health interventions have been implemented globally to control the spread of the virus. These measures have also had substantial impacts on the epidemiological patterns of other common respiratory viruses (8).

In this retrospective study, we analyzed the epidemiological characteristics of respiratory viral infections among pediatric patients treated at Shenzhen Children's Hospital from January 2020 to February 2025. Among the 13 respiratory pathogens tested, HRV, RSV, MP, and HPIV exhibited the highest positivity rates, at 24.57%, 12.46%, 11.46%, and 6.86%, respectively. These findings differ from previous reports—such as those by Lv et al. (9) in central Shandong, where the most prevalent pathogens were Streptococcus pneumoniae, HRV, RSV, and FluA, and by Ye et al. (10) in Zhejiang, where FluA, ADV, FluB, and RSV were predominant. This suggests that the distribution of respiratory pathogens among children may vary by region. Building on this observation, we further investigated the epidemiology of the four most prevalent pathogens, focusing on differences in infection characteristics before and after the onset of the COVID-19 pandemic, patterns of co-infection, gender-specific distributions, seasonal trends, and age-related differences among pediatric patients.

**TABLE 3** Detection rates of each ARVI pathogen in different pediatric age groups

| Age group | Pathogen | Total positive rate (%) | During the COVID-19 pandemic (%) | After the COVID-19 pandemic (%) | $\chi^2$ | P |
|---|---|---|---|---|---|---|
| 0–6 m | HRV | 19.48 | 18.94 | 20.37 | 1.044 | 0.3069 |
| | RSV | 24.54 | 24.21 | 25.32 | 1.832 | 0.1770 |
| | MP | 1.57 | 0.47 | 2.72 | 88.43 | <0.0001 |
| | HPIV | 6.59 | 5.25 | 8.56 | 46.96 | <0.0001 |
| 6–12 m | HRV | 27.38 | 24.28 | 28.36 | 21.23 | <0.0001 |
| | RSV | 20.68 | 19.78 | 21.02 | 5.413 | <0.05 |
| | MP | 4.12 | 1.31 | 6.84 | 124.2 | <0.0001 |
| | HPIV | 10.55 | 8.16 | 12.16 | 30.46 | <0.0001 |
| 1–3 y | HRV | 27.30 | 23.47 | 28.45 | 56.55 | <0.0001 |
| | RSV | 18.04 | 18.62 | 16.32 | 16.22 | <0.0001 |
| | MP | 7.91 | 1.97 | 12.75 | 706.3 | <0.0001 |
| | HPIV | 9.08 | 7.92 | 10.16 | 26.69 | < 0.0001 |
| 3–6 y | HRV | 28.59 | 26.31 | 29.02 | 16.94 | <0.0001 |
| | RSV | 8.00 | 8.13 | 6.91 | 10.30 | <0.01 |
| | MP | 14.54 | 3.3 | 21.31 | 1236 | <0.0001 |
| | HPIV | 7.02 | 7.28 | 6.62 | 3.220 | 0.0728 |
| 6–12 y | HRV | 20.43 | 17.75 | 21.59 | 26.89 | <0.0001 |
| | RSV | 2.05 | 1.69 | 2.12 | 0.3663 | 0.5450 |
| | MP | 24.15 | 5.97 | 31.55 | 1054 | <0.0001 |
| | HPIV | 2.73 | 2.19 | 2.85 | 5.05 | <0.05 |
| 12–18 y | HRV | 13.78 | 10.89 | 15.33 | 5.527 | <0.05 |
| | RSV | 1.67 | 1.81 | 1.57 | 0.498 | 0.8234 |
| | MP | 6.23 | 0.78 | 9.23 | 36.02 | <0.0001 |
| | HPIV | 2.77 | 1.16 | 3.47 | 7.067 | <0.01 |

During the COVID-19 pandemic, strict public health interventions significantly suppressed the transmission of multiple respiratory viruses, resulting in a markedly lower overall detection rate compared to the post-pandemic period. With the relaxation of control measures, the overall positive detection rate of respiratory viruses rose sharply from 56.22% during the pandemic to 71.04% post-pandemic, indicating intensified viral circulation—a trend consistent with reported rates of 70%–80% in the literature (11, 12). Notably, positivity rates for major pathogens such as HRV, RSV, MP, and HPIV differed significantly between the two stages: HRV (21.92%) and RSV (14.10%) predominated during the pandemic, while HRV (25.07%) and MP (17.71%) became the most prevalent after measures were relaxed. Coinfections, particularly triple or even quadruple ones, became much more common post-relaxation, with HRV most frequently involved, followed by RSV. Epidemiological studies from Saudi Arabia and regions such as Chongqing, China, demonstrate that non-pharmaceutical interventions (NPIs) interrupted the typical seasonality and transmission patterns of traditional pathogens like RSV, PIV-3, and influenza, sometimes causing these viruses to "disappear" or shift their epidemic peaks dramatically. After NPIs were lifted, these viruses rebounded with atypical, off-season surges or shifted peaks (13, 14). According to the "immunity debt" theory, suppressed viral circulation during the pandemic weakened population-level immunity, contributing to the post-pandemic upsurge and increased coinfection rates for RSV, influenza, and other pathogens (15). These changes highlight the need to dynamically adjust prevention strategies and vaccination schedules based on new epidemiological patterns.

This study systematically revealed that the epidemiological characteristics—seasonal, age, and gender-related—of various respiratory viruses changed significantly during and after the COVID-19 pandemic. In terms of seasonality, HPIV peaked in autumn and winter during the pandemic but shifted to a summer peak post-pandemic, indicating a marked change in seasonal pattern. HRV maintained strong seasonality with peaks in autumn and winter during both periods. MP was evenly distributed throughout the year during

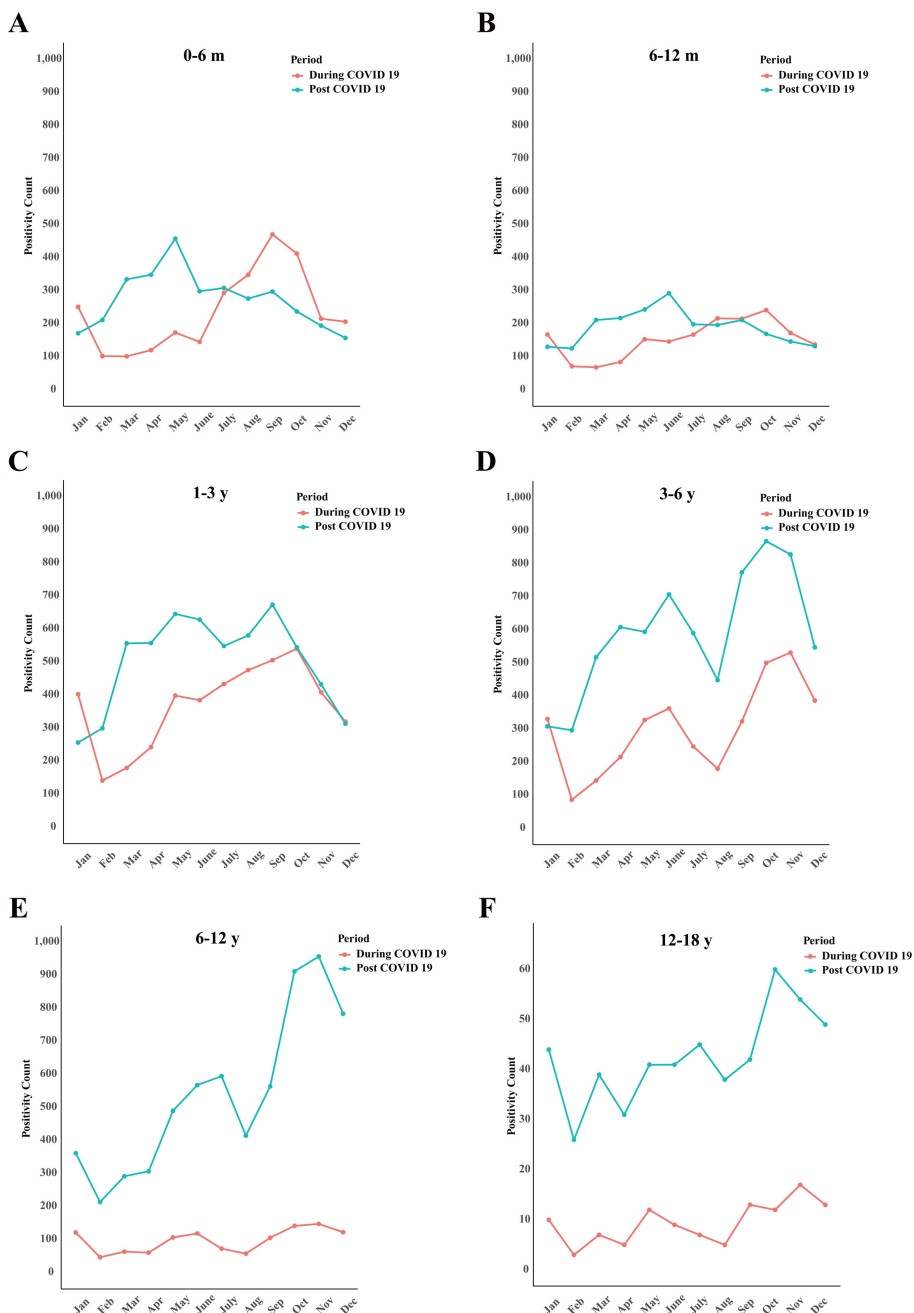

**FIG 3** Monthly comparison of respiratory virus specimen detection among different age groups during the COVID-19 pandemic (red) versus the post-pandemic period (blue). Patients were stratified into six age categories: 0–6 months (A), 6–12 months (B), 1–3 years (C), 3–6 years (D), 6–12 years (E), and 12–18 years (F). Trends are displayed monthly, highlighting differences in virus detection across age groups in the context of pandemic and post-pandemic settings.

the pandemic, with a slight decline in spring, but became more concentrated in summer and autumn afterwards. RSV shifted from a summer-autumn peak during the pandemic to a gradual decrease in positivity rate from spring to winter post-pandemic. These patterns are consistent with studies conducted in Saudi Arabia, China, and worldwide, all indicating that the relaxation of pandemic control measures led to staggered peaks and rebounds in respiratory virus activity, with some viruses (e.g., RSV and influenza) exhibiting atypical peaks and seasonality shifts due to "immunity debt" (16–18).

Regarding gender, the overall positive rate among males (67.03%) was significantly higher than that among females (65.41%), and the gender differences in positivity rates

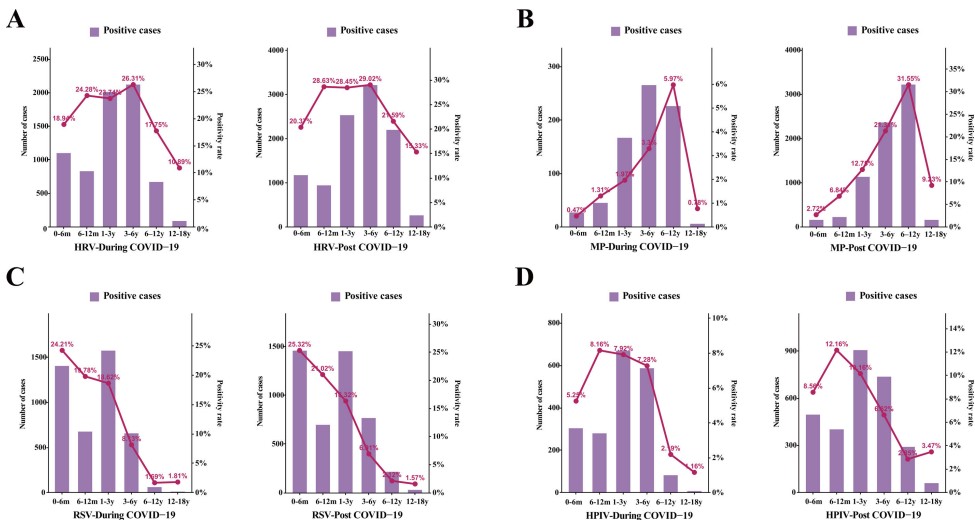

**FIG 4** Distribution of the number and proportion of virus-positive specimens among different age groups during the COVID-19 pandemic and the post-pandemic period, including HRV (A), MP (B), RSV (C), and HPIV (D).

for HRV, RSV, MP, and HPIV were statistically significant. After the pandemic, positivity rates for HRV, MP, and HPIV increased in both genders, while RSV showed a decrease. These findings are consistent with previous reports, suggesting that certain respiratory viruses display sex-based differences in susceptibility and transmission. The generally higher infection rate among male children may be related to factors such as immune system maturity and exposure opportunities (19–21).

Age stratification analysis revealed pronounced differences in viral infection positivity rates among five age groups. HPIV was the most prevalent in infants aged 6 months to 1 year, HRV was the most common in the 3–6 year age group, MP exhibited the highest positivity in children older than 6 years, and RSV was most prominent in infants aged 0–6 months. These age-specific distributions are consistent with previous literature, such as the finding that RSV predominantly infects infants younger than 6 months and remains the leading cause of lower respiratory tract infection in young children worldwide (17). Furthermore, the data indicate that with the relaxation of pandemic control measures, the overall level of viral infections increased markedly among all age groups, especially among preschool and school-aged children, with a rapid rise in the positivity rates of various respiratory viruses (22). This suggests that "immunity debt" may have profound implications for people of all ages, particularly young children and adolescents.

This study has several limitations. Most cases originated from Shenzhen, which may limit the representativeness and generalizability of the results. Future multicenter, large-scale studies are needed to validate these findings. The late introduction of multiplex respiratory pathogen testing in our hospital resulted in insufficient pre-pandemic data, restricting long-term trend comparisons. Furthermore, human rhinoviruses and enteroviruses share high genetic similarity, making them difficult to distinguish with standard molecular methods. This may lead to overestimation of rhinovirus prevalence and underestimation or misclassification of enterovirus infections, affecting detection specificity and introducing bias in regional epidemiological interpretation. Additionally, the "post-pandemic period" in this study (2023–2025) is not epidemiologically homogeneous. Changes in population mobility, school reopening schedules, interregional travel, and immunity levels in late 2024–2025 likely altered seasonal patterns and epidemic intensity compared with the early post-pandemic stage (2022–2023). This later phase, considered "post–post-pandemic," showed shifted pathogen peaks, seasonal restructuring, and more pronounced immunity debt effects, especially in certain age groups. Future surveillance and prevention strategies should distinguish between these phases and adopt refined, season- and age-specific interventions.

## Conclusion

Our findings demonstrate significant post-pandemic increases in respiratory virus positivity rates, with marked differences across pathogens, age groups, and genders. The seasonal distribution of HPIV, HRV, MP, and RSV shifted notably after COVID-19 restrictions eased. Increased co-infections and age- and gender-specific susceptibilities highlight the need for tailored prevention strategies in children.

## ACKNOWLEDGMENTS

This work was supported by Guangdong High-level Hospital Construction Fund.

Z.R. performed the conceptualization; T.W., and L.Z. conducted formal analysis; Y.C. acquired funding; Z.R. and T.W. carried out investigation; T.W., L.Z., and Z.R. developed methodology; Y.C. administered the project; L.Z. and T.W. handled software; T.W., and L.Z. performed validation; Z.R. created visualization; T.W. wrote the original draft; Z.R. reviewed and edited the manuscript.

## AUTHOR AFFILIATIONS

[1]Department of Laboratory Medicine, Shenzhen Children's Hospital, Shenzhen, China
[2]Department of Laboratory Medicine, Shenzhen Pediatrics Institute of Shantou University Medical College, Shenzhen, China

## AUTHOR ORCIDs

Tao Wu  http://orcid.org/0009-0003-4911-4949
Zhenmin Ren  http://orcid.org/0000-0001-7124-6882
Yunshen Chen  http://orcid.org/0000-0002-8520-9852

## AUTHOR CONTRIBUTIONS

Tao Wu, Formal analysis, Investigation, Methodology, Software, Validation, Writing – original draft | Liyang Zhong, Data curation, Software, Validation | Zhenmin Ren, Conceptualization, Data curation, Supervision, Validation | Yunshen Chen, Funding acquisition, Supervision

## ETHICS APPROVAL

This study was conducted in accordance with the Declaration of Helsinki and received approval from the Ethics Committee of Shenzhen Children's Hospital (Approval Number 202318002). The requirement for individual informed consent was waived by the ethics committee considering both the minimal risk to participants and the fact that the study had no impact on patient care or clinical outcomes.

## ADDITIONAL FILES

The following material is available online.

Open Peer Review

**PEER REVIEW HISTORY (review-history.pdf).** An accounting of the reviewer comments and feedback.

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
