## [Reviewer comments · Microbiology Spectrum]

Microbiology Spectrum

Epidemiological Shifts in Pediatric Respiratory Pathogens in Shenzhen, China: Impacts of COVID-19 Control Measures and Relaxation

Tao Wu, Liyang Zhong, Zhenmin Ren, and Yunshen Chen

Corresponding Author(s): Yunshen Chen, Shenzhen Children's Hospital

Review Timeline:

Submission Date:	August 15, 2025
Editorial Decision:	November 1, 2025
Revision Received:	November 27, 2025
Accepted:	December 16, 2025

Editor: Alexander Ivanov

Reviewer(s): Disclosure of reviewer identity is with reference to reviewer comments included in decision letter(s). The following individuals involved in review of your submission have agreed to reveal their identity: Karen K Kyuregyan (Reviewer #2)

Transaction Report:

DOI: <https://doi.org/10.1128/spectrum.02536-25>

Re: Spectrum02536-25 (Epidemiological Shifts in Pediatric Respiratory Pathogens in Shenzhen, China: Impacts of COVID-19 Control Measures and Relaxation)

Dear Dr. Yunshen Chen:

Thank you for the privilege of reviewing your work. Both reviewers noted that the study represents a technically correct retrospective study of respiratory viral infections in a dense region. However, they felt that the manuscript would benefit from revision and presented a list of questions and suggestions that you can find below, together with instructions from the Spectrum editorial office.

Revision Guidelines

Sincerely,
Alexander Ivanov
Editor
Microbiology Spectrum

Reviewer #1 (Comments for the Author):

The manuscript by Wu et al. presents nicely collected and analyzed data of ARTI causative agents circulation during (2020-2022) and after (2023-2025) the pandemic. Overall the presentation of the results is good and conclusions are sound. However, I'd like the authors to consider the following: the epi conditions and ARTI agents circulation right after the measures had been waived in 2022-2023 were not the same as in end-2024-2025, when most of the population have encountered ARTI

agents. It could be considered, probably, as the next post-pandemic period and be described separately.

Minor comment:

- Font on the Figures 1-4 is extremely small and hard to distinguish.

Reviewer #2 (Comments for the Author):

The paper by Wu and coauthors presents results of retrospective study of the detection rates of 13 ARVI pathogens in pediatric patients during and after COVID-19 pandemic. This study confirms the observation already made in different parts of the world that and nonpharmaceutical interventions and subsequent immunity gaps due to COVID-19 pandemic resulted in changes in ARVI pathogen prevalence rates and their seasonal patterns. The study is large-scale and executed well. However, there are some comments regarding study design and data presentation.

1. Lines 110-111. What was the reason behind the stratification of patients into following age groups: (1) 0-6 months; (2) 7-12 months; (3) 1-3 years; (4) 3-6 years; and (5) over 6 years. Why all patients 6-18 years were included in one group?
2. Sub-section 2.3. Please indicate if the assay used in the study was in house assay or commercially available kit. If the latter, please indicate the name of the kit and the manufacturer. If the assay was in house test, please provide data on its analytical performance.
3. The title of Sub-section 3.1 requires revision.
4. Table 2 contains data on detection rates of only 4 out of 13 ARVI pathogens during and after COVID-19. Please add data on other 7 ARVI pathogens and provide data on coinfections in separate table. It is recommended to merge data on detection rates of 13 ARVI pathogens in one table where one column contains data on overall detection rates, and two other columns - data during and after COVID-19.
5. Sub-section 3.2. Authors should clearly state why they analyzed seasonal patterns for only four viruses (HRV, MP, RSV and HPIV). Was it because positivity rates for other viruses during and after COVID-19 pandemic were similar? However, the seasonal patterns could be shifted even if the positivity rates were the same.
6. Sub-section 3.4. Please provide data on detection rates of each ARVI pathogen in different age groups as a Table with data presented as 1) overall positivity rate; 2) during pandemic; 3) after pandemic, with statistically significant differences indicated.
7. The grouping all children and adolescents from 6 to 18 years old into one age group does not seem correct. Please consider dividing this group into several age groups (for instance, 6-12, 13-18).
8. Line 232. FluAnts - infants?
9. Figure 4 is not clear. What do bars "Total cases" mean? Number of samples tested? If so, these data are excessive, data on number of positive cases (absolute numbers) and positivity rate values are enough.

Reviewer #1 (Comments for the Author):

The manuscript by Wu et al. presents nicely collected and analyzed data of ARTI causative agents circulation during (2020-2022) and after (2023-2025) the pandemic. Overall the presentation of the results is good and conclusions are sound.

However, I'd like the authors to consider the following: the epi conditions and ARTI agents circulation right after the measures had been waived in 2022-2023 were not the same as in end-2024-2025, when most of the population have encountered ARTI agents. It could be considered, probably, as the next post-post-pandemic period and be described separately.

R: We sincerely thank the reviewer for the insightful suggestion. We have carefully addressed this point by revising the Discussion section to distinguish between the early post-pandemic phase (2022-2023) and the later phase (2024-2025), which we refer to as the “post-post-pandemic” period. We now describe the differing epidemiological conditions and ARTI agents’ circulation patterns between these two phases, and we emphasize that future surveillance and prevention strategies should consider these phases separately with more refined, season- and age-specific interventions.

Minor comment:

- Font on the Figures 1-4 is extremely small and hard to distinguish.

R: We sincerely appreciate the reviewer’s valuable feedback regarding the font size in Figures 1-4. To improve clarity and readability, we have revised all figures by enlarging the font and optimizing the layout, ensuring that the text is now easier to distinguish.

Reviewer #2 (Comments for the Author):

The paper by Wu and coauthors presents results of retrospective study of the detection rates of 13 ARVI pathogens in pediatric patients during and after COVID-19 pandemic. This study confirms the observation already made in different parts of the world that and nonpharmaceutical interventions and subsequent immunity gaps due to COVID-19 pandemic resulted in changes in ARVI pathogen prevalence rates and their seasonal patterns. The study is large-scale and executed well. However, there are some comments regarding study design and data presentation.

1.Lines 110-111. What was the reason behind the stratification of patients into following age groups: (1) 0-6 months; (2) 7-12 months; (3) 1-3 years; (4) 3-6 years; and (5) over 6 years. Why all patients 6-18 years were included in one group?

R: We categorized patients into five age groups: 0 – 6 months, 7 – 12 months, 1 – 3 years, 3 – 6 years, and >6 years, primarily based on differences in immune system maturation, susceptibility to respiratory pathogens, and severity of clinical infections among infants and children. Younger age groups generally have less mature immune systems, resulting in higher infection risks and more severe disease outcomes; thus, this stratification facilitates comparative analysis of infection rates and clinical

characteristics across developmental stages. In our initial analysis, patients aged >6 years were treated as a single group; however, in subsequent statistical analyses, we further subdivided them into 6 – 12 years and 13 – 18 years to capture notable differences in immune experience, patterns of social contact, and exposure risks between school-age children and adolescents, thereby enabling the development of more targeted public health strategies and clinical prevention measures.

2. Sub-section 2.3. Please indicate if the assay used in the study was in house assay or commercially available kit. If the latter, please indicate the name of the kit and the manufacturer. If the assay was in house test, please provide data on its analytical performance.

R: In this study, detection of the 13 respiratory pathogens was performed using a commercial kit, namely the multiplex assay kit for 13 respiratory pathogens (fluorescence PCR-capillary electrophoresis method; Health Gene Technologies Co., Ltd., Ningbo, China), rather than an in-house developed method, and this information has been explicitly stated in the Methods section.

3. The title of Sub-section 3.1 requires revision.

R: We thank the reviewer for the constructive suggestion. In accordance with your advice, we have revised the title of Section 3.1 to "Positivity and Distribution of 13 Respiratory Pathogens", and this change has been implemented in the manuscript to more accurately reflect the content of the section .

4. Table 2 contains data on detection rates of only 4 out of 13 ARVI pathogens during and after COVID-19. Please add data on other 7 ARVI pathogens and provide data on coinfections in separate table. It is recommended to merge data on detection rates of 13 ARVI pathogens in one table where one column contains data on overall detection rates, and two other columns - data during and after COVID-19.

R: Thank you for your valuable comments. We have supplemented the detection rate data for the remaining nine respiratory pathogens and provided detailed results of mixed infections in a separate table. In addition, we have consolidated the detection rates of all 13 pathogens into a single table, including overall detection rates as well as separate columns for the COVID-19 pandemic period and the post-pandemic period. These additional data and the integrated table have been incorporated into the revised manuscript to facilitate a clearer comparison of pathogen prevalence across different stages and trends in mixed infections.

5. Sub-section 3.2. Authors should clearly state why they analyzed seasonal patterns for only four viruses (HRV, MP, RSV and HPIV). Was it because positivity rates for other viruses during and after COVID-19 pandemic were similar? However, the seasonal patterns could be shifted even if the positivity rates were the same.

R: We appreciate the reviewer's valuable comment. We focused our seasonal trend analysis on HRV, MP, RSV, and HPIV because these four pathogens showed the highest positivity rates among the 13 tested respiratory pathogens and demonstrated

statistically significant differences ($P < 0.0001$) between the pandemic and post-relaxation periods, indicating clear seasonal variability. In contrast, other pathogens exhibited relatively similar overall positivity rates between the two stages, with smaller changes in their seasonal distribution and insufficient data to support reliable statistical analysis. Selecting these high-prevalence pathogens ensured representative and statistically robust seasonal trend analyses, allowing us to more accurately capture the impact of public health policy changes on the epidemic seasonality of major respiratory pathogens.

6. Sub-section 3.4. Please provide data on detection rates of each ARVI pathogen in different age groups as a Table with data presented as 1) overall positivity rate; 2) during pandemic; 3) after pandemic, with statistically significant differences indicated.

R: Dear Reviewer, thank you for your valuable comments. In response to your suggestion, we have added data on the detection rates of each ARVI pathogen across different age groups, presented in a table that includes: (1) overall positivity rate; (2) positivity rate during the pandemic; and (3) positivity rate after the pandemic. Statistically significant differences have been indicated within the table.

7. The grouping all children and adolescents from 6 to 18 years old into one age group does not seem correct. Please consider dividing this group into several age groups (for instance, 6-12, 13-18).

R: Dear Reviewer, thank you very much for your valuable suggestion regarding age group classification. In accordance with your advice, we have subdivided the original 6-18 year age group into two separate groups: 6-12 years and 13-18 years, and have reanalyzed the data accordingly. The revised manuscript now includes updated age group results along with detailed statistical analyses, ensuring a more accurate representation of epidemiological characteristics across different age stages.

8. Line 232. FluAnts - infants?

R: We thank the reviewer for pointing out this typographical error. The term "FluAnts" in the manuscript should indeed be corrected to "Infants," and we have made this revision accordingly to ensure accuracy .

9. Figure 4 is not clear. What do bars "Total cases" mean? Number of samples tested? If so, these data are excessive, data on number of positive cases (absolute numbers) and positivity rate values are enough.

R: We appreciate the reviewer's valuable comments. In Figure 4, the "Total cases" bar indeed represents the total number of specimens tested. As this information is not essential in this context, we have removed it and retained only the number of positive cases and the positive rate, and have redrawn the figure accordingly.

Re: Spectrum02536-25R1 (Epidemiological Shifts in Pediatric Respiratory Pathogens in Shenzhen, China: Impacts of COVID-19 Control Measures and Relaxation)

Dear Dr. Yunshen Chen:

Your manuscript has been accepted, and I am forwarding it to the ASM production staff for publication. Your paper will first be checked to make sure all elements meet the technical requirements. ASM staff will contact you if anything needs to be revised before copyediting and production can begin. Otherwise, you will be notified when your proofs are ready to be viewed.

Sincerely,
Alexander Ivanov
Editor
Microbiology Spectrum